# Problematizing Child Maltreatment: Learning from New Zealand's Policies

Hamed Nazari [1,*], James C. Oleson [2] and Irene De Haan [3]

1   School of Social Sciences, University of Auckland, B201, 10 Symonds Street, Auckland 1010, New Zealand
2   School of Social Sciences, University of Auckland, 10 Symonds Street, Room 923, Private Bag 92019, Auckland 1142, New Zealand; j.oleson@auckland.ac.nz
3   School of Counselling, Human Services and Social Work, University of Auckland, B201, 10 Symonds Street, Auckland 1010, New Zealand; i.dehaan@auckland.ac.nz
*   Correspondence: hnaz030@aucklanduni.ac.nz

**Abstract:** Since all policies address problems, they necessarily include implicit or explicit constructions of these problems. This paper explores how child maltreatment has been constructed in New Zealand's child protection policies. It questions the underlying assumptions of this problem construction and seeks to shed light on what has been omitted. Utilizing a qualitative content analysis of eight key policy documents, this study reveals the construction of child maltreatment has been dominated primarily by a child-centric, risk-focused approach. This approach assigns blame and shifts responsibilities onto parents and families. In addition, the vulnerability discourse and social investment approach underpinning this perspective have allowed important structural factors, such as poverty and inequality, to remain unaddressed. This paper also highlights the one-dimensional focus on the lower social class to control future liabilities. We suggest that the harm inflicted by corporations on children's well-being is another form of child exploitation currently omitted from the problem construction. We suggest that child abuse should be defined and understood in policy as harm to children's well-being and argue that the state should prevent and mitigate harm by addressing structural forces of the problem as well as protecting children against corporate harms.

**Keywords:** child maltreatment; child protection; child well-being; social investment; materialism; corporations

## 1. Introduction

In 2007, the United Nations International Children's Emergency Fund (UNICEF) published a report about six dimensions of child well-being, including material well-being, health and safety, educational well-being, family and peer relationships, behaviors and risks, and subjective well-being, across 30 developed countries (Adamson 2007). According to the report, New Zealand's performance was below average in children's material well-being and health and safety, with the latter being the second worst (after the United States) among the 25 countries assessed. In 2020, UNICEF updated the child well-being index for 38 European Union (EU) and Organization for Economic Cooperation and Development (OECD) countries, and found that New Zealand's poor performance in child well-being has continued over the past decade, with similar trends observed, especially in terms of physical health and mental well-being (UNICEF 2020). According to the 2020 report, New Zealand ranks 33 in the physical health index, 38 in mental well-being, and 23 in academic and social skills among 38 other EU and OECD countries.

Suicidal ideation, high among New Zealand secondary school students (Fleming et al. 2007), was one of the indicators of mental well-being in the UNICEF report. Studies show that there is strong link between child abuse exposure and suicide attempts, suicide plans, and suicidal ideation (Afifi et al. 2016). Research also shows that child maltreatment has consequences for different aspects of well-being. It can affect the physical and mental health of

children (e.g., Davidson et al. 2009; Irish et al. 2010), increase the likelihood of risky behaviors (e.g., Boden et al. 2009), and affect social skills as well as the educational performance of children (e.g., Boden et al. 2007).

To safeguard children against abusive behaviors, all developed countries have established child protection policies (Gilbert et al. 2011). Like other developed countries, concerns about child abuse in New Zealand increased in the 1970s and 1980s after mass child removal, especially from Indigenous families. There is a consensus among scholars that modern child protection in New Zealand began with the enactment of the Children, Young Persons, and their Families (CYP&F) Act in 1989 (Hyslop 2017). The core purpose of this Act was to include family involvement in decision making about providing care and support for children. It has been argued that the New Zealand child welfare system found a family service orientation after the CYP&F Act (Gilbert et al. 2011; Gilbert et al. 2012); however, other social policy changes over the 1990s, including the reduction in social protections such as income protection, alongside unaffordable housing and cuts to the funding of non-governmental service providers, led to high child poverty rates and ultimately an increase in the rates of notifications to child protection services in the 2000s (Keddell and Davie 2018).

During the 2008–2017 leadership of the National Party, the 2010s marked a significant shift in child protection policies. This administration sought to reform New Zealand's child protection legislation and related policies, enacting 2014's Vulnerable Children Act and issuing the 2012 White Paper for Vulnerable Children (the White Paper, hereafter) and the 2015 Modernising Child, Youth and Family report. In addition, to restructure child protection practices, a new ministry was established in 2017: the Ministry for Children (Oranga Tamariki). This new ministry focuses on five aspects of child protection: prevention, intensive intervention, care services, transition, and youth justice (Keddell and Davie 2018). However, despite years of policy development and implementation, New Zealand still ranks poorly among developed nations in terms of child well-being (UNICEF 2020).

Alongside the emergence of the new policy direction in social policy, scholars have increasingly turned their attention to the multifaceted landscape of child protection in New Zealand. The existing literature outlines three aspects of child abuse and protection. First, descriptions of the development of the child protection system. For example, Fernandez and Atwool (2013) provide an insightful overview of the early history of care and protection in Australia and New Zealand, while Hyslop (2013, 2017, 2022) provides a comprehensive overview of the political history of New Zealand's child protection system. Second, scholars have explored the views of child abuse survivors, including, for example, on the protective apparatus, as evidenced by Jülich (2006), who asked survivors of abuse in New Zealand what they thought about the state's protective mechanisms, while Proietti-Scifoni and Daly (2011) sought the views of survivors of historical child abuse in order to scrutinize the appropriateness of restorative justice for addressing partner, family, and sexual violence, alongside child sexual abuse. Third, critical analyses of New Zealand's child protection systems. For example, Hackell (2016) criticized the government's neoliberal social policy discourse, while other studies addressed the impact of racism, inequality, and neoliberalism on the country's child welfare policies and practices (e.g., Hyslop 2017; Hyslop and Keddell 2018; Keddell and Davie 2018). Moreover, the association between child maltreatment and deprivation is underscored by Keddell et al.'s (2019) findings, revealing marked differences in the likelihood of child abuse cases, family group conferences, and foster care placements based on socio-economic disparities.

While a significant amount of research has been conducted on child maltreatment in New Zealand, limited attention has been given to child protection policy. To the best of our knowledge, there is no systematic study that examines the construction of child abuse as a problem in child protection policy documents. Therefore, this study is designed to understand how child maltreatment is constructed in New Zealand's child protection policies. As Bacchi (2009) explains, there exists an underlying assumption that policies are inherently beneficial, with the belief that they serve to solve problems. This implies the

existence of a problem, even if this problem is not explicitly detailed. Instead, it is implicit in the very essence of policymaking, as policies are designed to instigate change, signifying the presence of underlying issues. This is where the "What's the problem represented to be?" (WPR) perspective becomes relevant. The WPR perspective argues for the significance of bringing the implicit problems inherent in public policies to the surface and subjecting them to careful examination. Therefore, it can be said that all problems in social policies have a constructive nature, meaning they are presented and understood in a certain way. Drawing on Bacchi's (2009) WPR approach for critical policy analysis, our study seeks to answer three linked questions:

- How is child maltreatment *constructed* in the child protection policies?
- What *assumptions* underlie the construction of child maltreatment?
- What is *omitted* from the construction of this problem?

The first two questions are answered in the Section 3 of this article and the third question is dealt with in the Section 4.

## 2. Method and Materials

### 2.1. Qualitative Content Analysis

Our data collection and analysis employed a qualitative content analysis method. Unlike quantitative content analysis, which is based on procedures such as "frequency analysis", "analyses of indicators", and "analysis of intensity", qualitative content analysis delves deeper into the content of written and verbal data to discover "latent meaning" (Flick 2014; Flick et al. 2004). Qualitative content analysis does not limit itself to manifest content and frequency counts; rather, it aims to go beyond the words and interpret the latent meaning of materials. Drawing on Roller and Lavrakas (2015) and Schreier (2012), qualitative content analysis was utilized in this study as a method to systematically summarize the content, to identify themes, and to extract interpretations of both the manifest meanings (the visible and obvious components) and the latent meanings (interpretation of the underlying assumptions) of the data.

We followed Lincoln and Guba (1985), Graneheim et al. (2017), and Schreier (2012) in conducting an inductive content analysis, taking a data-driven approach to the identification of patterns in the manifest content and latent meaning. So, this study utilizes qualitative content analysis to analyze New Zealand's child protection policy documents and describe and interpret how child maltreatment and its causes are constructed. The researchers' primary goal is to summarize the content of policy documents, extract themes and categories, and interpret the findings.

We thought carefully about reflexivity and the role that our own biases played in the research. Even though our research involved content analysis of policy documents rather than something immediately, palpably human (such as interviews or ethnography), the framing of our research questions and the analytical steps used to answer them were nevertheless influenced by our training and experience. As Bourke (2014) notes, "To achieve a pure objectivism is a naïve quest, and we can never truly divorce ourselves of subjectivity. We can strive to remain objective, but must be ever mindful of our subjectivities. Such is positionality" (p. 3). The lead author of this study is a doctoral student in criminology with a background in research methods, especially qualitative methods; the second author is a lawyer–criminologist with experience in criminal law policy; the third author is a registered social worker with experience in New Zealand's Office of the Chief Social Worker. All three authors are immigrants to New Zealand: from Iran, the USA, and Scotland. And it is likely that this insider–outsider status has allowed us to note some aspects of child maltreatment policy while overlooking other features. As a function of our collective experience, we have identified eight key policy documents as seminally important, while excluding a whole range of other documents as peripheral or duplicative. In doing so, of course, we remain subject to a range of biases (confirmation bias, availability bias, and selection bias, among others)—a fact that acknowledging our positionality does not itself remedy (Savolainen et al. 2023)—but we hope that being reflexive about how we ask and how we answer our

research questions allows us to have critical conversations and, ultimately, to perform better social research on child maltreatment and child protection policy.

In this research, the term "content" refers to the child protection policy documents of New Zealand. The term "policy document" also refers to reports that are directives or guidelines established by governments to regulate child protection as well as laws enacted by a legislative body. In addition, the term "discourse" in this article refers to the understanding of a particular social problem, its causes, and how to react to/resolve it.

### 2.2. Data Collection

To collect relevant content, the initial phase involved selecting the appropriate materials, a task undertaken following a thorough literature review. In total, eight documents, listed below, underwent scrutiny in this study:

- Puao-te-Ata-tu (daybreak), 1988 (Māori Perspective Advisory Committee 1988).
- The 1989 Children, Young Persons and their Families Act, amended in 2017 as the Oranga Tamariki Act (Oranga Tamariki Act 2017).
- The 2012 White Paper for Vulnerable Children (Volumes I and II) (Ministry of Social Development 2012a, 2012b).
- The 2014 Children's Act (Children's Act 2014).
- The 2015 Modernising Child, Youth and Family report, 2015 (Interim and Final reports) (Expert Panel_Modernising Child Youth and Family 2015a, 2015b).
- The 2019 Child and Youth Well-being Strategy (DPMC 2019).

Figure 1 highlights the changes in the New Zealand political ecosystem and the timing of the reviewed documents.

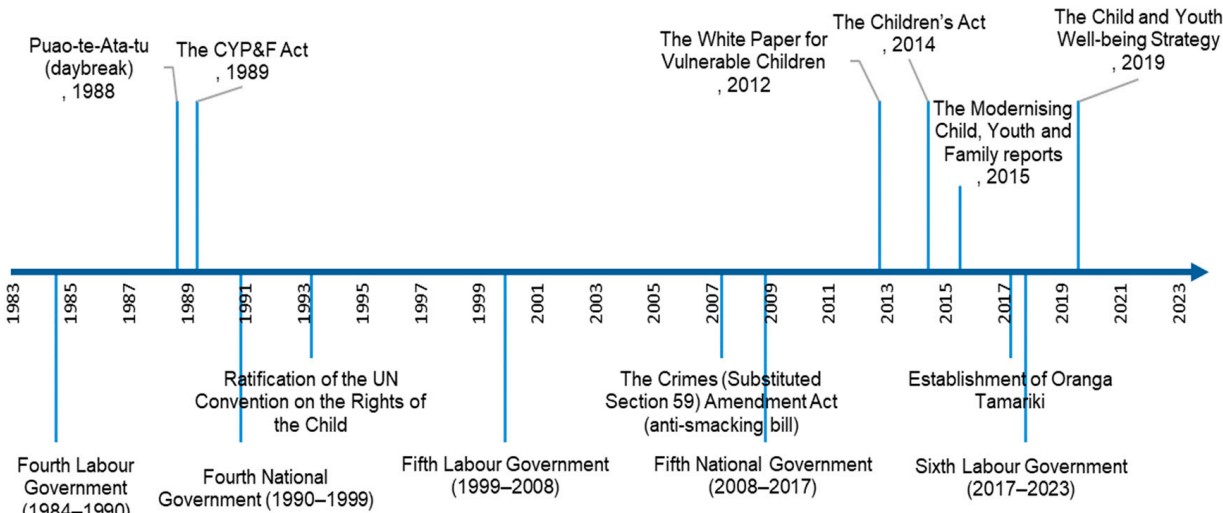

**Figure 1.** Timeline of reviewed documents and contemporary government.

These documents were selected because they represent the most significant changes in the child protection policy and practice in New Zealand. For instance, Puao-te-Ata-tu (daybreak) stands as a pioneering document that challenged the state's role in child removal and that shifted attention toward issues of institutional racism and discrimination (Māori Perspective Advisory Committee 1988). It also became the cornerstone of the CYP&F Act in 1989, an Act that shared the state's decision-making power with families. Applying the social investment logic to child protection has also been prototyped in the changes made under the fifth National Government (the White Paper, the Children's Act, and the Modernising Child, Youth, and Family reports). The Child and Youth Well-being Strategy was also chosen for inclusion in this analysis as it represents the shift in policy direction under the sixth Labour Government (2017–2023). It is also worth noting that

while these documents are the most impactful ones in terms of policy direction, there are other documents that were not analyzed in this research.

Building on Clapton's (2022) work, upon a close read of the documents, it became evident that there have been notable shifts in both the length and language utilized in child protection documents over time. For instance, Puao-te-Ata-tu, as the cornerstone of the CYP&F Act, comprised 92 pages and presented 13 recommendations aimed at reforming social welfare policy. A lexical analysis of this document shows that the most frequent keywords (with their stemmed words) were 'Māori', occurring 649 times; 'people', 205 times; 'welfare', 195 times; 'community', 156 times; and 'culture', 139 times. However, documents associated with reforms in the 2010s exhibited an expansion in length. The White Paper (Volume I and II combined) spanned 222 pages. Keywords recurrent in the White Paper included 'child', occurring 3254 times; 'family', 955 times; 'service', 644 times; 'vulnerable', 544 times; 'parent', 458 times; and 'development', 415 times. The Modernising Child, Youth, and Family reports (Interim and Final reports combined)—aimed at reforming child protection practice in New Zealand—extended to 457 pages. Main keywords in these reports comprised 'child', repeated 4117 times; 'service', 1260 times; 'support', 1113 times; 'investment', 705 times; 'vulnerable', 598 times; and 'future', 575 times. Despite its brevity at 92 pages, the Child and Youth Wellbeing Strategy shares thematic similarities with the previous documents, with 'child' recurring 747 times, 'wellbeing' 344 times, 'support' 219 times, 'family' 147 times, and 'development' 139 times.

### 2.3. Data Analysis

The coding process began with reading the selected documents and identifying meaningful units, which were then highlighted. Reflective notes were taken to capture the researchers' questions, ideas, and perspectives. All materials were then coded through open and axial coding processes (Flick 2014). Open coding constituted the first step, where text was broken down into meaningful units and assigned conceptual labels that represent the meaning of each unit (see Table 1 for examples of open coding). This was an inductive process that involved creating data-driven summaries from the data.

**Table 1.** Examples of data extracts and coding.

| Category | Theme | Code (Example) | Retrieved Section (Example) |
|---|---|---|---|
| From socio-cultural context to individualization of child maltreatment | Racism and historical roots | Monocultural laws and administration → Racism as the root cause | It is our view that the presence of racism in the Department is a reflection of racism which exists generally within the community. Institutional racism exists within the Department as it does generally through our national institutional structures. Its effects in this case are monocultural laws and administration in child and family welfare, social security or other departmental responsibilities. Whether or not intended, it gives rise to practices which are discriminatory against Māori people. (Māori Perspective Advisory Committee 1988, p. 24) |
| | | Impact of colonization | Throughout colonial history, inappropriate structures and Pakeha involvement in issues critical for Maori have worked to break down traditional Maori society by weakening its base-the whanau, the hapu, the iwi. It has been almost impossible for Maori to maintain tribal responsibility for their own people (Māori Perspective Advisory Committee 1988, p. 18) |

**Table 1.** *Cont.*

| Category | Theme | Code (Example) | Retrieved Section (Example) |
|---|---|---|---|
| From socio-cultural context to individualization of child maltreatment | Individualization and family responsibility | Family responsibility | The White Paper outlines a set of reforms that help to ensure that parents, caregivers, family, whānau and communities understand and fulfil their responsibilities towards children, as the single most critical factor in the care and protection of vulnerable children. (Ministry of Social Development 2012a, p. 2) |
| | | Incompetent parents | Parents exert the most profound influence over the development of their children, for good or ill. Good parenting provides a protective environment for the early years, providing positive experiences that boost healthy brain development and a protective cocoon against sources of stress and harm. The vast majority of parents wish to do their best for their children, although not all have the knowledge, skills and resources to meet their development needs. A small minority does not have their best interests primarily in mind. There is a vast literature on the many ways that parents affect their children's development. Much of the complexity in this literature can be summarised by focusing on two key topics: parent-child attachment and authoritative parenting. (Ministry of Social Development 2012a, p. 15) |
| | Getting tough on abusers | Longer sentences | Amending the Crimes Act in 2011 to: <br> - Introduce a new offence that means any member of a household with frequent contact with a child and who knows a child is at risk from abuse but fails to take reasonable steps to protect the child may be prosecuted. <br> - Broaden the scope of the duties of parents and those with actual responsibility for children. These people will be held liable if they fail to take reasonable steps to protect a child from injury. Thoughtlessness or ignorance is no longer a defence, and penalties for ill-treatment or neglect of a child have been doubled to a maximum of 10 years' imprisonment. (Ministry of Social Development 2012b, p. 27) |
| | | Surveillance | Information about at-risk children or families will be logged in a new information system and, if further action is needed, information will be accessed by relevant professionals so they can see the whole picture for a child. If a vulnerable child is referred to a community provider, a child and family worker, or government agency, all relevant information on the child will be available to them so they have the facts at their fingertips. (Ministry of Social Development 2012b, p. 7) |

**Table 1.** *Cont.*

| Category | Theme | Code (Example) | Retrieved Section (Example) |
|---|---|---|---|
| From socio-cultural context to individualization of child maltreatment | High-risk groups | Monitoring high-risk adults | The Government will extend and systematise existing arrangements for monitoring high-risk adults to include those subject to the proposed Child Abuse Prevention Order and other groups of high-risk adults, and to ensure that relevant information remains accessible over time. (Ministry of Social Development 2012b, p. 20) |
| | | Child abuse prevention orders | New child abuse prevention orders (civil orders) will allow a judge to place restrictions in situations where an individual poses a high risk to a child or children in the future. Restrictions may include taking action to advise the parents or caregiver of the child and to remove the high-risk adult from the situation if appropriate. Another potential restriction is that when a child has been removed from a home due to serious child abuse at the hands of a parent, the existence of a child abuse prevention order could mean that another baby born into that situation is removed from that parent's care. (Ministry of Social Development 2012b, p. 20) |
| | Safe and loving home | Providing loving home as the most important responsibility | Ensuring we love, care and nurture all our children and young people throughout their lives is the most important task we have. This Strategy is our collective call to action. (DPMC 2019, p. 3) |
| | | Good life for children | a good life for children and young people means being loved, happy, supported by their family and friends, and being connected to their whānau, communities, languages and cultures. Children and young people want to be accepted for who they are, listened to, and supported in their aspirations. (DPMC 2019, p. 11) |
| Social investment approach | Applying social investment approach | Child as human capital | A good start in life helps children to experience the best of childhood. The children of today are also the parents, workers and business and community leaders of tomorrow. To ensure future economic and social success, it is important that children are healthy, well nurtured and well educated so they are well equipped to assume these future roles. Investment in children can reduce the emergence of problems that have high social and fiscal costs. (Ministry of Social Development 2012a, p. 39) |
| | | Early identification as goal → Multiagency approach as tool → Social investment as foundation | The investment approach for vulnerable children will use data, evidence and analytics to: <br>- provide incentives to intervene with the right service, as early as practicable with the right children, young people and families, by ensuring that agencies and non-government providers are accountable for achieving improved outcomes which will reduce costs in the longer term; <br>- identify, evaluate and improve on the interventions; and <br>- create a shared and consistent prioritisation and delivery of seamless services for children, young people and families across multiple agencies and providers within the sector. (Expert Panel_Modernising Child Youth and Family 2015b, p. 16) |

**Table 1.** *Cont.*

| Category | Theme | Code (Example) | Retrieved Section (Example) |
|---|---|---|---|
| Social investment approach | Vulnerability and vulnerable children | Focus on early identification of vulnerable children | Recent advances in research and technology mean we can now start to get ahead of the problem, identifying and helping some 20,000–30,000 vulnerable children and families, in many cases before the greatest harm occurs. (Ministry of Social Development 2012b, p. 5) |
| | | Continuous focus on vulnerable children | In order to achieve greater equity, the Government has prioritised policies and initiatives to improve the wellbeing of children and young people who are living in poverty and disadvantaged circumstances, those of interest to Oranga Tamariki, and those with mental health or additional learning needs. (DPMC 2019, p. 60) |
| | Risk assessment and early identification | Risk assessment by using big data | The Government will build new tools to help professionals identify which children are most at risk. A risk predictor tool has been tested by the Ministry of Social Development in conjunction with the University of Auckland and will be further developed to include a wider range of data. Using available information held by government, it has been possible to develop statistical criteria that help identify children at greater risk, based on information about them and their family circumstances. This approach could be used to guide professionals towards those children who are most vulnerable. (Ministry of Social Development 2012b, p. 10) |
| | | Early intervention | Making sure that the right needs and risks are responded to at the right level, by identifying and intervening earlier with vulnerable children and their families/whānau who need intensive family support. (Ministry of Social Development 2012a, p. 115) |
| | Child-centric service | Placing children at the center | The system does not place children at the centre. The current operating model is focused on process rather than on the needs of the child. Current legislation does not consistently support a child centred approach. (Expert Panel_Modernising Child Youth and Family 2015b, p. 15) |
| | | Shift from service to child-centered | The system must shift from being primarily centred on the services, processes and administrative convenience of the agencies, to bringing the voice of children, young people and their families to the forefront. (Expert Panel_Modernising Child Youth and Family 2015b, p. 10) |

The next step was axial coding, which involved extracting themes from the available concepts, refining and distinguishing existing concepts, and summarizing them into categories. This involved a deductive process through which the researchers created themes (Mayring 2014), recombining the data broken down by open coding in order to identify relationships between them (Flick 2014; Flick et al. 2004). This process was facilitated by using NVivo as the data analysis tool.

## 3. Results

### 3.1. Individualization of Child Maltreatment in the Child Protection Policy

To understand how the problem of child maltreatment and its root cause have been constructed within child protection policies, it is essential to scrutinize the definition and

conceptualization of child maltreatment. Thus, the objective of this section is to explain what is deemed problematic by examining changes proposed within the policy.

*Puao-te-Ata-tu* constructed child maltreatment as a socio-cultural problem that has three underlying causes. First, to understand child maltreatment and its overrepresentation among Māori [the Indigenous people of New Zealand], the historical impact of colonization must be acknowledged. This includes recognizing the loss of land, culture, and language, as well as the impact of discriminatory policies and practices. Secondly, from a legal perspective, child abuse was connected to the dominance of the colonial Western worldview in social policy and demonstrated ignorance of Māori ways of childrearing and family structure in law. The report specifically focused on the placement of Māori children and notes that the placement was based on a colonial legal framework that was not designed to meet the needs of Māori (e.g., see Jackson 1987 on how legal systems have affected indigenous people). Thirdly, child abuse as a social problem was related to institutional racism. This includes biases within government agencies, such as police and child welfare services, that disadvantage Māori and other minority groups (Māori Perspective Advisory Committee 1988).

It can be argued that child maltreatment in *Puao-te-Ata-tu* was represented as fundamentally rooted in socio-cultural factors, particularly the disparity in 'child-rearing practice' and family structure between Māori and Pākehā [New Zealanders of European descent]. *Puao-te-Ata-tu* highlighted that the primary issue lies in the miscomprehension of the role of children in Māori culture and their connections to whānau [Māori families/extended families], hapū [Māori sub-tribe], and iwi [Māori tribe] structures. The core message of the report is that there exists "a profound misunderstanding or ignorance of the place of the child in Māori society and its relationship with whānau, hapū, and iwi structures" (Māori Perspective Advisory Committee 1988, p. 7). So, at the heart of this problem is the fundamental contrast between Māori child-rearing practices and those of colonial Western culture. In Māori culture, placing children with whānau or hapū members served to fortify kinship structures. The family was not merely a nuclear unit but an integral part of the larger tribal community, united by reciprocal obligations. Older family members, unburdened by daily demands, often served as the best caregivers for children due to their blood ties and ongoing contact with the child's birth family (Māori Perspective Advisory Committee 1988). According to *Puao-te-Ata-tu*, the issue within the legal system was that "decision-making and responsibility for the placement of children has been taken from close-knit Māori communities and the authority placed exclusively in courts and state agencies" (Māori Perspective Advisory Committee 1988, p. 97). The primary objective of the CYP&F Act, 1989, was essentially to share this decision-making power with the community.

The call for addressing the socio-cultural context of child maltreatment in *Puao-te-Ata-tu* resulted in the enactment of the Children, Young Persons, and their Families Act (CYP&F Act) in 1989 (renamed to Oranga Tamariki in 2017) (Oranga Tamariki Act 2017). The CYP&F Act pursued dual objectives: preserving family and whānau stability through the establishment and maintenance of responsive community-based support and promoting shared decision-making authority to create partnerships between families and official authorities (Marsh 1994). Central to this approach is the Family Group Conference (FGC) model, which has been described as "New Zealand's gift to the world" (Becroft 2017). The FGC mandates a meeting involving the child or young person, their family (including extended family and primary caregivers), and relevant professionals before crucial decisions regarding children are made (Levine 2000; Marsh 1994).

With the initiation of the social investment approach in New Zealand during the period 2011–12 under the fifth National Government and applying investment logic (using resources today with the expectation of yielding future benefits) to the government's social interventions (Boston and Gill 2017), the vulnerability discourse became dominant as a new model for the child protection system. This new discourse was represented by the White Paper for Vulnerable Children:

the Government will focus on vulnerable children in a way that has never been done before. The White Paper for Vulnerable Children sets out a programme of change that will shine a light on abuse, neglect, and harm by identifying our most vulnerable children and targeting services to them . . . . (Ministry of Social Development 2012b, p. 4)

The main function of the vulnerability discourse was to identify and control risk that may cause future liabilities (Hyslop 2022; Stanley and de Froideville 2020). The core notion in the White Paper was that vulnerable children should be identified as early as possible and effective interventions made to prevent future harm and cost. Therefore, the first question was 'who are vulnerable children?'. According to the White Paper,

vulnerable children are children who are at significant risk of harm to their well-being now and into the future as a consequence of *the environment in which they are being raised* and, in some cases, due to their own complex needs. Environmental factors that influence child vulnerability include not having their basic emotional, physical, social, developmental and/or cultural needs met at home or in their wider community. (Ministry of Social Development 2012b, p. 6; *italics added*)

Three years later, in 2015, the Modernising Child, Youth and Family report presented an even narrower definition of vulnerable children:

vulnerable children are those who are at significant risk of harm now and in the future as a consequence of *their family environment* and/or their complex needs, as well as those who have offended or may offend in the future. (Expert Panel_Modernising Child Youth and Family 2015b, p. 41; *italics added*)

The revised conceptualization of child abuse as the vulnerability of specific groups of individuals and the dysfunction within certain family environments resulted in the individualization of the problem, meaning the root causes of child abuse were ascribed to the characteristics of individuals and/or families:

the White Paper outlines a set of reforms that help to ensure that parents, care-givers, family, whānau and communities understand and fulfil their responsibilities towards children, *as the single most critical factor* in the care and protection of vulnerable children. (Ministry of Social Development 2012a, p. 2; *italics added*)

Hyslop (2022) asserts that neoliberal subjectivity rests on the notion that individuals must invest in themselves as marketable commodities in order to attain self-responsible well-being. In such a context, the core concern is to promote personal responsibility and accountability, thereby making families take responsibility for their economic and social circumstances. As highlighted in the White Paper,

a working breadwinner is the best form of security a family can get . . . Jobs bring financial and social rewards, building family strength and pride . . . Recent welfare reforms which focus on helping more people back to work, with the financial and social benefits that brings, will play a big part in preventing vulnerability . . . Programmes that support children to achieve in education, such as those delivered through the Youth Guarantee, the new Youth Service, and enhanced education and training opportunities, *combine to get young people ready for work and help grow our economy* . . . Though income poverty alone does not cause or excuse child abuse, we know that struggling to make ends meet places extra stress on families . . . *We also know that it is what parents do that matters most*. (Ministry of Social Development 2012b, p. 26, *italics added*)

As a result of this new understanding of the problem of child abuse, the government's efforts are targeted towards either rescuing or providing service to at-risk children and families rather than addressing the socio-economic factors of social problems (Stanley and de Froideville 2020). This is in line with a medical understanding of the problem. As Parton (1985) notes, from a medical perspective, child maltreatment is seen as a symptom of a

disorder, disease, or syndrome which can be explained by the perpetrator's personality or family background. Consequently, efforts are made to change them through a treatment programme to ensure that the problem does not recur in the future. Despite ample evidence linking child abuse to poverty and other socio-structural factors, the White Paper simply put emphasis on psychological aspects of parenting, such as 'parent–child attachment', 'positive parenting', and 'parents' knowledge about child development'. Defining the problem in this way places blame on parents and elevates the role of parental psychology and denies the role of social structural factors (Ferguson 2004; Hackell 2016; Hyslop 2017).

Additionally, the reduction in the root causes of child abuse to individual behaviors has led to the predominance of punitive responses. If interventions targeting individual or familial pathologies prove ineffective, the inclination is towards adopting stricter measures against abusers. The White Paper proposes new measures for the surveillance and management of high-risk individuals, as follows:

> amending the Crimes Act in 2011 to: . . . broaden the scope of the duties of parents and those with actual responsibility for children. These people will be held liable if they fail to take reasonable steps to protect a child from injury. Thoughtlessness or ignorance is no longer a defence, and penalties for ill-treatment or neglect of a child have been doubled to a maximum of 10 years' imprisonment. (Ministry of Social Development 2012b, p. 27)

> the Government will extend and systematise existing arrangements for monitoring high-risk adults to include those subject to the proposed Child Abuse Prevention Order and other groups of high-risk adults, and to ensure that relevant information remains accessible over time. (Ministry of Social Development 2012b, p. 20)

In 2014, the Vulnerable Children Act (Children's Act 2014) was enacted to provide the legal framework for the new discourse. The Modernising Child, Youth and Family reports were also published to outline the child protection practice required (Expert Panel_Modernising Child Youth and Family 2015a), and a new Ministry for Vulnerable Children (renamed to Ministry for Children later in 2017) was established in 2017 to provide services for the core populations of interest. As a result, the entire system's focus underwent a transformation to better align with the discourse surrounding risk and vulnerability. Nevertheless, in 2019, the sixth Labour government introduced a new strategy for the public sector called the Child and Youth Well-being Strategy (the Strategy, hereafter) that outlines three key priorities for the government to address the challenges faced by New Zealand's children and young people: (a) tackling family and sexual violence; (b) reducing child poverty; and (c) improving mental health and well-being (DPMC 2019).

At a first glance, it might seem that the system's objective has undergone substantial revision, expanding its services to encompass the structural factors contributing to child maltreatment. However, upon closer scrutiny of the Strategy's content, it becomes apparent that, firstly, the emphasis persists on addressing the needs of those with the most pressing needs (vulnerable children). Secondly, the primary aim of the system is to support families by providing *safe, stable, and loving homes* for children, aligning closely with the mantra of the Modernising Child, Youth and Family report:

> the best place for a child is in the *safe, loving and stable care of their families*, whānau, hapū, iwi or other family group. A stable and quality home environment with love and trust influences a child and young person's wellbeing every day, and their ability to form attachments to others. (DPMC 2019, p. 35; *italics added*)

The Strategy acknowledges that most children in New Zealand are doing well, while underscoring the objective of enhancing the lives of those who are not. It appears that despite the absence of the term 'vulnerable' in the Strategy, the government's focus remained on improving outcomes for the most vulnerable children. In the last section of the Strategy, where the government's priorities are outlined, this direction becomes even clearer. The government commits to adopting a strategy to improve the well-being of *children*

*with greater needs*. This indicates a targeted approach towards those children who require additional support and assistance due to their specific circumstances or vulnerabilities.

In sum, whether the policy direction is aimed towards removing children from abusive environments and placing them in state care or supporting parents to take care of their children by teaching them parenting skills and providing them with financial and health support to create *safe, stable, and loving homes* for children, the point is that we have connected the problem of child maltreatment to individual parents who cannot take care of their children and child abuse is constructed as a behavior of an individual that needs to be corrected. Essentially, the idea of focusing on the home environment as a solution to child abuse is considered by some scholars (e.g., Hyslop 2022) as overly simplistic. "Within a neoliberal capitalist society characterized by the production and reproduction of deeply classed, gendered and raced inequalities, this prescription is akin to mowing the grass and expecting it not to grow again" (Hyslop 2022, p. 151). This is not to say we do not need to work with individual children to make them safe from an abuser. The problem is that this formulation of the problem in policy blames families for the harm inflicted by an inequitable system. Put briefly, while the problem of child maltreatment was linked to the structural factors in the early documents, individual behaviors and family characteristics are constructed as the source of the problem in the latest policy documents.

### *3.2. Understanding Child Protection Policy in the Context of the Social Investment Approach*

Gilbert et al. (2011) conducted an analysis of 10 developed countries and identified two distinct orientations to child protection: child protection and family service. The two orientations differed in four key aspects. Firstly, they diverged in their fundamental perception of child abuse: one viewed it as a need to protect children from 'degenerative relatives', while the other saw it as a product of family conflict and dysfunction, amenable to assistance. Secondly, this led to distinct response mechanisms, with one being legalistic and the other service-oriented, focusing on therapeutic interventions and needs assessment. Thirdly, child welfare professionals played contrasting roles, either adversarial in the child protection orientation or collaborative, particularly with parents, in the family service orientation. Lastly, with regard to out-of-home placements, the family service orientation preferred voluntary arrangements with parents, while the child protection orientation relied on state coercion, often through court orders, to enforce such placements. Then, drawing on the classic work of Esping-Andersen (1990), Gilbert and colleagues clustered countries into three groups: liberal (Anglo-American), conservative (Continental), and social democratic (Nordic). Their analysis suggested that Anglo-American nations are oriented towards child protection as their primary approach, whereas Continental European and Nordic countries adopt a family service-oriented approach when dealing with maltreatment issues while differing in their policies regarding mandatory reporting.

It appears that child protection in the *Puao-te-Ata-tu* report was aligned with a blended model of the Nordic and Continental European approaches. However, the subsequent publication of the White Paper resulted in a reframing of the system's objectives, which were then more closely aligned with the Anglo-American model. Hyslop (2022) argues that this shift is represented by the White Paper. He describes this shift as "the simplistic and politically populist position of getting tough on child abusers" (p. 119), indicating a significant and deliberate change in the underlying principles and objectives of the child protection system in New Zealand. Hyslop also suggests that the Vulnerable Children Act 2014, as the legal framework of the White Paper, primarily focuses on safeguarding children through the strengthening of measures aimed at identifying those who perpetrate abuse. So, one could argue that after the White Paper, New Zealand's child protection system moved closer to the Anglo-American model, emphasizing the idea of rescuing children from abusive environments. However, there is a subtle difference with the past in terms of the justifications underlying policy and practice. While in the past, children were removed from so-called amoral wicked caregivers who inflicted cruelty upon their innocent children

(Parton 1985), in the contemporary context, an economic rationale embedded in the social investment approach underpins the policy and practice.

Boston and Gill (2017) assert that the social investment approach has three aspects: investment in human capital, social protection, and actuarial risk assessment. Child protection in New Zealand has adopted a distinctly child-centric approach. This shift stems from the perception of children as human capital. Parton (2014) argues that child welfare systems have transformed from a historical dichotomy of 'child protection' and 'family service' to a contemporary discourse that overwhelmingly prioritizes children's developmental needs. This discourse, he contends, has evolved under the influence of the concept of a child as 'human capital', particularly within neoliberal political economies. Within this discourse, children are regarded as pivotal to future economic prosperity, envisioned as the 'citizen-workers of the future'. As stated in the White Paper,

> the children of today are also the parents, workers and business and community leaders of tomorrow. To ensure future economic and social success, it is important that children are healthy, well nurtured and well educated so they are well equipped to assume these future roles. Investment in children can reduce the emergence of problems that have high social and fiscal costs. (Ministry of Social Development 2012a, p. 39)

Here, the second aspect of social investment, social protection, becomes relevant. It entails 'placing children at the center' and the provision of services and support essential for safeguarding the mental and physical well-being of children. The White Paper indicates that "we are proudly putting children at the centre of the picture—wrapping services and supports around them and their needs" (Ministry of Social Development 2012b, p. 6). This approach is rooted in the belief that policies promoting children's growth and development will ultimately yield sustained economic success for the nation, especially in the context of global competition. Consequently, a deliberate and targeted strategy is in place to maximize the potential of children, all in the pursuit of ensuring long-term national economic prosperity (Ridge 2012).

Ultimately, the triangle becomes completed by an actuarial risk assessment approach. To target the service to the right person, there is a need for assessing the risk of child maltreatment, which is conducted through actuarial methods. It was easy for the government to identify those children who had already experienced maltreatment because the government had access to the children's information. However, the challenge was in defining some factors to identify those who are 'at risk' of abuse. There were five types of risk factors in the White Paper that were used for identifying vulnerable children: (1) risk factors in the child (such as unwanted or high-needs child), (2) risk factors in parents and caregivers (such as abuse of alcohol or drugs, maltreated as a child, chronic mental or physical problems), (3) relationship factors (such as lack of parent–child attachment, family breakdown), (4) community factors (such as poor housing, transient neighborhoods), (5) societal factors (values that diminish status of child/parents, acceptance of violence and abuse). The White Paper suggested the use of 'big data' and predictive risk modelling (PRM) tools to help professionals identify children at risk of abuse or neglect and implement early intervention strategies (Ministry of Social Development 2012a; Vaithianathan et al. 2012). Based on these five risk factors and the ones that were included in the PRM model, it can be argued that the White Paper constructs child maltreatment as a problem of poor people exhibiting high-risk behaviors (e.g., drug and alcohol abuse) and a lack of parenting skills.

It also appears that the underlying assumptions of the Strategy have some commonalities with the White Paper and Modernising Child, Youth and Family report. This can be shown in two ways. The Strategy's development is closely aligned with the underlying model of the White Paper. Both documents propose an ecological approach to child protection, positioning the child at the center, surrounded by some potential risk factors. Secondly, the main focus of the strategy is on providing *safe, stable, and loving homes* for children. It seems that the family and home environment continue to be the central focus of the policy and practice in the Strategy. This is also highlighted in the statements made by Jacinda Ardern,

the Prime Minister at the time, that "ensuring we love, care and nurture all our children and young people throughout their lives is the most important task we have" (DPMC 2019, p. 3). The emphasis on providing 'safe, stable, and loving homes' for children at the earliest opportunity aligns with the stance promoted in the Modernising Child, Youth and Family report. This report emphasized the significance of creating supportive and nurturing family environments as a means of addressing the complex issue of child abuse and neglect (Expert Panel_Modernising Child Youth and Family 2015a). This is not to say that the Strategy focuses solely on the home environment. The Strategy acknowledges the importance of societal factors such as poverty. However, while the Strategy introduces new terminology and moves away from explicitly using the term 'vulnerable child', it does share similarities with the previous discourse in terms of being child-centric, targeting welfare provision, emphasizing early intervention, and highlighting the importance of parental responsibilities.

## 4. Discussion and Conclusions

### 4.1. Discussion

Undoubtedly, conventional forms of child abuse cause harm to children's well-being. However, as Bacchi (2009) invites us to reflect on issues silenced in identified problem representations, this section addresses harms to children's well-being that are currently omitted from the construction of the problem of child maltreatment.

Although child abuse—defined by the United Nations (UN) (UNICEF 1989) as comprising physical, sexual, emotional/psychological, or neglect—is more prevalent among the lower social class (May-Chahal and Cawson 2005; Pelton 1978), other forms of behavior, currently not recognized as abuse, might also be detrimental to the well-being of children and might also impose costs upon society. For example, some scholars have discussed the potential adverse impacts of consumerism and materialism on individuals' well-being and behaviors (e.g., de Graaf et al. 2014; James 2007; Levine 2009). Levine (2009) argues that economic privilege comes at a price: materialism. Materialism, more prevalent among the upper social class, has the potential to erode the sense of purpose and altruism in children and young individuals, making them more self-centered and less concerned about the needs of others. This perspective finds resonance with other scholars who have similarly pointed out the dark side of affluence. Affluent people tend to place an excessive emphasis on values such as individualism, perfectionism, achievement, competition, and material wealth, often at the expense of nurturing prosocial values like empathy, cooperation, and altruism. This can lead to the cultivation of self-centeredness and a lack of concern for the well-being of others among children raised in such environments (de Graaf et al. 2014; Hamilton and Denniss 2005; James 2007).

Child rearing practice in affluent families can also lead to consequences and costs for society. In a pioneering study, Piff et al. (2012) demonstrated that a higher social class is predictive of unethical behavior. While it was previously assumed that individuals from lower socio-economic backgrounds might exhibit more unethical behavior due to their exposure to environments characterized by limited resources, heightened threats, and increased uncertainty (similar to Messner and Rosenfeld's (2013), in "Crime and the American Dream" theory of anomie), Piff's findings intriguingly revealed the contrary. Individuals from the upper class were shown to be more inclined towards unethical behavior, attributable to their access to greater resources, enhanced personal freedom, and reduced dependence on others, fostering self-centered social cognition and behavior. Moreover, those from the upper class showed lower levels of empathy, were worse at understanding others' emotions, and exhibited reduced levels of generosity and altruism (see also Piff 2014; Piff et al. 2010).

According to Piff et al. (2012), individuals from higher social classes are more likely to engage in behaviors such as cheating, fraud, and deception, which carry significant consequences for society. They are more inclined to break traffic regulations, make ethically questionable decisions, take valuable possessions from others, lie during negotiations, engage in dishonest tactics to enhance their chances of winning prizes, and endorse unethi-

cal conduct within a workplace setting. Several other studies also show that individuals from higher social classes are more likely to prioritise their personal well-being over the well-being of others. Consequently, they may exhibit more favorable attitudes towards greed, which is a robust determinant of unethical conduct (Kraus et al. 2009, 2010; Piff et al. 2010, 2012). Therefore, a higher social class predicts committing unethical conduct, reflecting a self-centered disposition.

Interestingly, research also indicates that people from higher social classes are more likely to be promoted to leadership roles within their organizations and they may have received education with an economic focus (Adler et al. 2000). Moreover, Babiak and Hare (2009) emphasize that psychopathic individuals (those "without conscience and incapable of empathy, guilt, or loyalty to anyone but themselves" (p. 19)) can be charming, manipulative, and often rise to positions of power within organizations. They also argue that psychopathic individuals in corporate settings can be more prone to committing white-collar crimes, such as fraud and embezzlement.

As argued, child abuse has been internationally defined as specific behaviors that cause harm to the physical and psychological well-being of children (UNICEF 1989). The focus on psychological harm has centered on conditions such as depression, anxiety, PTSD, and various mental disorders (e.g., Breslau et al. 2014; Moffitt et al. 2007; Scott et al. 2012). Notably absent from this discourse are considerations of traits like narcissism, psychopathy, and self-centeredness. This omission may be attributed to the alignment of these traits with colonial Western economic values, where capitalism thrives on the productivity of healthy citizen-workers. Those experiencing depression, anxiety, and stress might struggle to actively contribute to economic growth, whereas self-centered individuals, motivated by competition and manipulation rather than cooperation, may be better positioned for success (for a discussion about competition versus cooperation, refer to Harcourt 2021, 2023). Therefore, if we understand personhood as having value only as a wealth producer, raising materialistic self-centered children is not abuse but preparation for later successes. However, if we understand personhood as tied to community, to ethics, to non-economic relational bonds with families, religious communities, and civic organizations (for a discussion about the impacts of social disconnectedness, refer to Putnam 2000), then socializing young people to be unempathetic, narcissistic consumers might be abuse.

Another aspect of the problem of child abuse that is currently omitted from policy is corporations' harm to children's well-being. It was once believed that even if corporations do commit extremely expensive white-collar crimes, at least they were not causing harm to anyone. However, this notion has been proven to be a myth (Cullen et al. 2006). Having a direct impact on individuals' health and well-being, big businesses pollute air and water (Jackson 2016), market unhealthy food (targeting children, in particular) (Bakan 2011), and exploit workers by placing them in unsafe conditions (Chan et al. 2020).

In his book and film, *The Corporation*, Bakan (2003, 2005) argues that for-profit corporations act solely in their own self-interests. They are essentially programmed to prioritise wealth creation for their owners above all else. Following this mission leads to perceiving everything, including nature, human beings, children, and the planet, as opportunities to exploit for profit. He suggests that corporations, much like human psychopaths, lack the capacity for genuine concern for others, to feel guilt and remorse for their actions, and to have a sense of moral obligation to obey laws and social conventions.

The resemblance of corporations to psychopaths (and the associated concerns regarding their singular focus on profit maximization) invite reflections on the broader societal implications. This prioritisation of financial gains over considerations for the well-being of individuals, communities, and the environment raises particular concerns for vulnerable groups such as children. In the pursuit of profit, corporations prioritise their own self-interest over children's interests. If in the past, child exploitation was confined to child labor, today, corporations can freely exploit children's forming and turbulent emotions and bodies. Bakan (2011), in another thought-provoking book, *Childhood under Siege*, asserts that

big businesses cause harm to different aspects of children's well-being, from endangering children's physical health to inflicting psychological and social harm.

The expanding industry of so-called 'kid marketing' specifically targets children and teenagers with products, contents, and services uniquely designed for them. Technological progress has allowed marketers to target children directly through television. The agenda, therefore, was to understand what children wanted (regardless of what their parents thought) and to design products to satisfy their wants. Today, this strategy has remained stable as the foundation of content production in the targeting of children in social media (Bakan 2011).

Relying on psychology and behavioral science, kid marketing entices children's fundamental emotions at the deepest level. Love, fear, mastery and independence, and humor are all targeted in television, video and online games, and social media to make children consume products that are designed for them (Lindstrom et al. 2003). As a result of pursuing this strategy, addiction has become a source of profitability for the gaming industry (Kent 2010; Niman 2014). Children commit suicide, experience starvation and exhaustion, and suffer from psychiatric disorders as a result of sitting and playing for excessive hours and withdraw from family and friends (Andreassen et al. 2016; Bakan 2011). Social media also reinforces obsessive engagement in a different way, where teenagers derive identity and status primarily from being online. The anxiety children and teenagers report when they are without cell phones reflects a fear of losing ones' self in online relationships. This leads to isolation from other aspects of life and fosters compulsive and narcissistic behaviors (Bakan 2011).

Another harmful aspect of kid marketing to children's mental health is what Bakan (2011) describes as 'prescriptions for profit'. Bakan provides extensive evidence that the increasing number of children being diagnosed by mental disorders cannot simply be explained by the myth that a greater number of children are becoming mentally ill or the discovery of more complicated methods for detecting childhood problems. Instead, one needs to understand the pharmaceutical industry's growing influence over medical science and practice (see also Cohen 2016). Child psychiatry has become an immensely profitable industry and, as a result, different types of medications with life-long side effects are produced and prescribed too quickly and too frequently for children. Moreover, big businesses cause physical harm to children with unhealthy products. According to Bakan (2011), marketing and advertising 'junk food' to children is now a billion-dollar industry. A growing body of evidence also suggest that exposure to industrial chemicals and pollutants (dumped in the environment by corporations) is connected to children's chronic health problems such as asthma, cancer, and autism (Landrigan and Garg 2002; Vrijheida et al. 2016; Wilson and Horrocks 2008). In addition, studies show that gasoline lead exposure is strongly associated with lowered IQ level in children and subsequent aggressive and delinquent behaviors (Carpenter and Nevin 2010; Nevin 2000). This is especially important in the context of New Zealand, where international reports constantly warn of the country's weak performance in child physical and mental health (e.g., Adamson 2007; UNICEF 2020).

The kid marketing industry's spectacular success over the past 50 years has been due in part to increasing technological progress. Today, computers, smartphones, and social media make marketers capable of reaching and dazzling children in a breadth and depth that was unimaginable in the 1960s, when magazines and comic books gave way to television (Bakan 2011). Several studies depict how technological progress become great obstacles to our freedom (e.g., see Harvey 2003; Oleson 2007, 2023). This is especially the case for children and teenagers, as social media fuels narcissism in young people and makes them uncapable of developing empathetic and other social skills they need to get through life (Turkle 2011). However, despite all of the harm that kid marketing inflicts on children, the detrimental effects of their activities have never been identified as a form of child exploitation or risk to the physical and mental health of children. Hence, it has been excluded from child protection policies.

*4.2. Conclusions*

In this study, we attempted to explore how child maltreatment has been constructed as a problem in New Zealand child protection policies, the fundamental assumptions about causes and consequences of the problem, as well as to problematize aspects of the problem that have been omitted.

Child maltreatment has increasingly been portrayed as a "personal trouble" rather than a "social issue" (Mills 1959), detached from its broader social context and root causes. The problem is often framed as a characteristic of morally depraved individuals deserving tough punishment or as a family dysfunction necessitating therapeutic intervention. Several scholars (e.g., Hackell 2016; Hyslop 2017; Hyslop and Keddell 2018; O'Brien 2016) have argued that changes in the child protection policy have been consistent with the neoliberalism project to situate both causes and solutions of child abuse and neglect within individuals and their immediate environment. Consequently, the blame, burden, and responsibility of child maltreatment have shifted towards parents, particularly those at the bottom of the social ladder, who have been labelled as incompetent. This has led to the ignorance of the socio-structural factors of child maltreatment (Hyslop 2022).

Socio-structural factors consist of both socio-cultural and socio-economic elements. After the *Puao-te-Ata-tu* report, the importance of socio-cultural factors has been underlined in several studies. For instance, Wilson (2016) emphasized the impact of colonization on whānau and how Eurocentric perceptions of child-rearing replaced traditional values and practices and the collective obligations and responsibilities held by whānau and hapū members. A study by Pihama et al. (2019) has also shown that the roots of family violence are in the 'historical trauma' experienced by Māori whānau and the colonial ideologies that changed Māori beliefs and ways of living. The ethnic bias in child protection practice in New Zealand (Keddell and Hyslop 2019) and the intergenerational transmission of child maltreatment (Schelbe and Geiger 2017) are other cultural factors that have been discussed in different studies. Therefore, one could argue that there have been some efforts made to address these socio-cultural factors (Hyslop 2022). However, there still exists a significant dismissal of socio-economic factors of child abuse. This is despite compelling evidence demonstrating the impact of poverty and deprivation on child abuse. For instance, Keddell et al. (2019), in a study on child protection inequalities in New Zealand, highlighted a relationship between deprivation and contact with the child protection system. The absence of a sociological lens in child protection policy and the dismissal of structural factors and inequalities that contribute to child abuse have also been observed in other countries (Bywaters 2013; Bywaters et al. 2015; Featherstone et al. 2019).

It was also argued that the way that child maltreatment is problematized sits within a broader context of neoliberalism and social investment approach to social policy. Neoliberalism's logic is naturally tailored to the adoption of cost-effective behavioral interventions (Hyslop 2017). As mentioned, the social investment framework is overtly ideological, incorporating the punitive approach that has historically plagued liberal politics within a neoliberal economic context. It operates as a discourse that is presented as common sense (resources should be targeted towards the riskiest individuals), while simultaneously concealing the economic drivers of inequality (the exploitative nature of capitalist economics and poverty as a problem resulting from social structures rather than poor individual choices) (Hyslop 2022).

In addition, child protection policies tend to problematize specific violent behaviors that impact children's well-being while overlooking other significant aspects of the issue. Residing in a deprived area plus belonging to a particular ethnicity can place you in the category of "risky families" once a neighbor reports a moment of frustration, such as shouting or spanking in response to a child's excessive engagement with video games or social media. In contrast, if you are affluent, raising a self-centered child, the same behavior would not raise an eyebrow. Furthermore, if your child experiences anxiety, depression, or behavioral disorders due to exposure to video games and social media, the blame falls on you as an incompetent parent rather than upon the multinational corporations that

commodify childhood and deliberately target your child's deepest emotions and affections. Social policy places the burden on parents for factors largely beyond their control, such as socio-economic circumstances and the content of media (in a broad sense). As argued, this imbalance is neither just nor fair.

**Author Contributions:** Conceptualization, H.N., J.C.O. and I.D.H.; methodology, H.N., J.C.O. and I.D.H.; software, H.N.; validation, H.N., J.C.O. and I.D.H.; formal analysis, H.N.; investigation, H.N., J.C.O. and I.D.H.; resources, H.N., J.C.O. and I.D.H.; data curation, H.N.; writing—original draft preparation, H.N.; writing—review and editing, H.N., J.C.O. and I.D.H.; visualization, H.N.; supervision, J.C.O. and I.D.H.; project administration, H.N. All authors have read and agreed to the published version of the manuscript.

**Funding:** This research received no external funding.

**Institutional Review Board Statement:** Not applicable.

**Informed Consent Statement:** Not applicable.

**Data Availability Statement:** Publicly available policy documents were analyzed in this study. These data can be found here: https://www.legislation.govt.nz/, accessed on 6 January 2023 and https://www.msd.govt.nz/, accessed on 6 January 2023 and https://www.orangatamariki.govt.nz/, accessed on 9 January 2023.

**Conflicts of Interest:** The authors declare no potential conflicts of interest with respect to the research, authorship, and/or publication of this article.

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
