# Peer review of "Problematizing Child Maltreatment: Learning from New Zealand’s Policies"

_socsci, doi:10.3390/socsci13040193_

Round 1
Reviewer 1 Report
Comments and Suggestions for Authors
This is an interesting paper that as noted, focuses upon a little-examined dimension of child protection -the making of policy.
I have rated it in need of a little more work because the latter half of the paper tells us much of what we already know about society's relationship to children and, crucially moves off the subject of the policy papers. The latter, the policy papers, in my view are not given the kind of close reading that would enrich the claims and arguments made in the paper as a whole. Clapton's paper on Scottish child protection policy might be looked at for a close reading example, eg scrutiny of language, frequency of terms, burgeoning of lists, noting increase in word length of successive policy documents and so on (Clapton 2022, 'Beyond Intention. The Draft National Guidance for Child Protection in Scotland (2020): A Case Study of a Scottish Policy Document', Scottish Affairs).
So more use of the content of the documents would improve the paper, plus where appropriate, some contextualisation of them eg the given reason for a particular document's appearance might be interrogated.
Plus, more minor point, all chunks of quoted text ought to be introduced with a preceding colon.
I look forward to reading a revised version of the paper.
Comments on the Quality of English Language
Excellent.
Reviewer 2 Report
Comments and Suggestions for Authors
Thank you for the opportunity to review this article, which comes from a very interesting idea of how the conceptual understanding of the problem of child maltreatment is framed in policy documents. I was really interested to read your work and see the results. Your article is well written (if a little long and dense in style) but I think there are some fundamental issues that need addressed before it is suitable for publication. These concerns are:
1. The researcher/s have not positioned themselves, and given that this is a qualitative and largely theoretical piece of work, I think this is crucial. The arguments made in the work come from an absolutely subjective lens, which is absolutely fine as long as you say what that positioning is. All interpretation has a perspective and you need to clarify yours - otherwise it appears that you think that this is objective material, which it isn't. Not helping is the use of terms like 'in fact', and that there is particular 'proof' in particular arguments. Along with providing position statements and linking your beliefs and experiences to the work, I would recommend a close edit for use of stylistic and grammatical features that suggest that this there is anything beyond evidence for a position.
2 A timeline that shows the changes in the Aotearoa/NZ political ecosystem and the timing of the reviewed documents and how they correspond would be a good way of presenting the historical information and be helpful for readers from other countries.
- The methodology section and description needs attention - there is no search strategy so it isn't clear why you focused on these particular documents, apart from the fact that they are high profile. The analysis is also not clear, not assisted by the example in table 1, which isn't explained in the text. It isn't clear to me how the codes, concepts, categories and themes relate to each other. It is noted that there were 137 open codes and 9 themes but it isn't clear how these were derived or relate to each other. It isn't stated what the 9 themes are and it also isn't clear how these relate to the 5 sections of the results. And, a personal bugbear of mine - the use of the objective verb '9 themes were extracted from the data' as if they existed there within the data and could be objectively plucked out. similar to the point above, it's important that you own your part in the analysis process.
3 Your argument is largely founded on ideas about policy being influenced by a particular discourse and a paradigm. You need to define what you mean by each of these terms. You've reviewed policy documents but some are legislation and at times what you're describing feels like organisational and structural development or processes rather than policy. Having a clear description of how you understand policy will clarify this. I am also not sure that there is such a thing as a 'social investment paradigm' per se so I need convincing of this by appropriate academic means (citations, definitions etc). It is also important that you link these concepts more closely with the data - in what ways did you see these reflected in the data?
3 There is considerable crossover and replication in the results section and again in the conclusion - the arguments are repeated in different ways and need more consolidation and refinement. The article is too long and needs a close edit.
- The main issue that needs addressed is that some of the primary arguments in your article are not supported by any reference to your data. 3.3,4 and 5 all refer primarily to other research or review documents and not to the policy documents you analysed in this work. There is also a serious flaw in your discussion in 3.4 - essentially the argument you make is that middle class parents abuse their children in different ways from 'lower class' parents if abuse is defined differently - this is not the same thing as your title or conclusion for the section (that policy problematises lower class parenting practices). Likewise with section 3.5 - I have no doubt that much of what you argue is correct but there is no evidence that supports this argument in your data and, in fact, you don't even reference any of the documents in this 'result'. Just because other people or corporations 'abuse' children in different ways, it doesn't mean that child maltreatment in the conventional way of thinking about it doesn't exist. I think your point is that these things are absent from your data but that's a discussion point not results. The evidence for both of these arguments is also very selective and because it isn't related to your research findings, it feels like 'cherry picking' evidence to support your pre-existing views.
I hope that you can address these issues and refine your article to clearly link policy documents with particular understandings of child maltreatment in a more coherent way because I think it'll be a valuable contribution to the thinking around the topic if you can consolidate your thinking to focus on that. It maybe that your work in 3.4 and 3.5 are part of the discussion/implications if the arguments can be made more systematic and reasoned (and linked to your findings)
Round 2
Reviewer 1 Report
Comments and Suggestions for Authors
All suggested modifications actioned.